# Study on the Influence of Water Content on Mechanical Properties and Acoustic Emission Characteristics of Sandstone: Case Study from China Based on a Sandstone from the Nanyang Area

**Xin Huang** [1], **Tong Wang** [1], **Yanbin Luo** [2,*] and **Jiaqi Guo** [1,2,*]

1 School of Civil Engineering, Henan Polytechnic University, Jiaozuo 454000, China
2 School of Highway, Chang'an University, Xi'an 710064, China
* Correspondence: lyb@chd.edu.cn (Y.L.); gjq519@163.com (J.G.)

**Abstract:** In order to study the influence of water content on the mechanical properties of sandstone and evolution of crack propagation, laboratory compression tests and Engineering Discrete Element Method (EDEM) numerical simulation of sandstone under different conditions were carried out by the RMT-150B rock mechanics test system. The sandstone samples were from Nanyang, Henan Province, containing a total of 12 rock samples. Under the confining pressure of 0, 5, 10, and 20 MPa, the rock samples with 0%, 1.81%, and 3.24% water content were tested. The findings demonstrated that as the sample's water content grew, the peak strain increased but the peak strength, elastic modulus, maximum energy rate of individual acoustic emission events, and cumulative acoustic emission energy rate all reduced. While the ratio of tensile cracks to shear cracks inside the rock samples rose with increasing water content, the failure mode of sandstone changes from shear failure to tensile failure with the increase of water content, but the sandstone specimens in the three conditions exhibited shear macroscopic fracture surfaces. Research results will provide reference for the safe construction of underground projects in water rich areas.

**Keywords:** sandstone; water content; mechanical properties; acoustic emission; failure mode; EDEM

## 1. Introduction

In recent years, with the rapid development of underground engineering in China, tunnel collapse, water inrush, mud inrush, and other major engineering disasters caused by water rock interaction have been increasing, and the influence of water on rock stability has gradually been paid attention to [1]. Water content and surrounding rock pressure have a great impact on rock strength and failure deformation. In addition to softening the rock, water also dissolves some of its mineral constituents, enlarging the rock's interior fractures and altering its mechanical properties; this is especially true for sandstone, a rock with high porosity [2–4]. Therefore, the research on the influence of water content on the mechanical properties, failure characteristics, and internal damage evolution of rocks will provide references for the safe construction of underground projects in water-rich areas [5].

Domestic and foreign scholars have done a lot of experimental research on the impact of water on rock failure and have achieved fruitful results. K. Hashiba et al. [6] used uniaxial compression test to research the alternating loading rate of Mitsui Andesite under varying levels of water saturation, and the findings demonstrated that rock's uniaxial compressive strength rises as water saturation falls. Zhao et al. [7] conducted uniaxial compression test on red sandstone with different water content and compared the results by discrete element software. It was found that the strength and elastic modulus of rock decreased with the increase of water content. Feng et al. [8] carried out uniaxial compression tests of coal samples with different water content. Through the analysis of test results, they found that water softened the coal samples while weakening their brittleness, and the failure process

was more moderate. Eunhye Kim [9] conducted static and rapid loading compression tests on sandstone with different water contents; rock strength and Young's modulus decrease with the increase of water content. Sun et al. [10] carried out triaxial loading tests of sandstone rock mass with different water content, analyzed the acoustic emission signals inside the rock during loading, and found that with the increase of moisture content, the strength of the rock gradually decreased, and the elasticity and accumulated damage of the rock decreased, resulting in the gradual decrease of the accumulated acoustic emission count and accumulated energy in the process of broken. Erguler Z [11] tested different types of rock and found that the average elastic modulus and tensile strength decreased significantly as the UCS water content decreased. Guo et al. [12] conducted uniaxial compression and conventional triaxial compression tests on limestone under different moisture content, and discovered that when water content increased, limestone's uniaxial compressive strength declined and its peak strain increased. Tan et al. [13] used PFC software to simulate the cyclic loading and unloading of coal samples under different confining pressures. By analyzing the change law of elastic modulus and the evolution law of plastic strain of coal samples during loading, they proposed that the increase of confining pressure is conducive to the suppression of damage and the increase of the sample's ability to withstand plastic strain.

Domestic and foreign scholars have made a lot of achievements in the study of the influence of water content on the mechanical properties and failure modes of rocks. However, the accuracy of the indoor test depends not only on the process of the test, but also on the monitoring means adopted in the test to a large extent. However, the monitoring of most studies is unilateral, and few studies have observed and analyzed the internal evolution process of rocks with different water contents during loading. In order to study the influence of water content on the mechanical properties of sandstone and evolution of crack propagation from many aspects. In this paper, a series of compression tests are carried out on sandstone samples with different water contents, and the acoustic emission system is used to monitor the process of broken rock and with the aid of engineering discrete element method (EDEM), simulation with uniaxial and triaxial compression tests of sandstone with different water content, and analyzes the damage evolution process of rock samples at the microcosmic level. Combined with the results of laboratory tests and numerical tests, water content has a significant negative effect on sandstone strength and will lead to the evolution of internal cracks from shear to tensile. The test results reveal the influence of water content on the mechanical properties, deformation characteristics, and failure characteristics of sandstone.

## 2. Materials and Methods

### 2.1. Sample Preparation

Sandstone samples are taken from Nanyang, Henan Province, China. The color is gray white and the appearance is rough. They are composed of various sand cementation. The rock mass is relatively complete and there are no obvious defects on the surface. Sampling and sample preparation are carried out in strict accordance with GB/T2356.1-2009 "rock physical and mechanical properties determination method", machining of field rock blocks into standard cylindrical specimens with a diameter of 50 mm and a height of 100 mm [14]. Before the test, the sound velocity of the obtained rock samples is calibrated, and the individuals with large wave speed deviation eliminated so as to minimize the influence of rock discreteness on the experimental results. The prepared standard samples were divided into three groups: dry group, natural group, and saturated group, respectively labeled D, N, and S. Compression tests were carried out with confining pressures of 0 MPa, 5 MPa, 10 MPa, and 20 MPa. Different confining pressures of dry group, natural group, and saturated group were D-0, D-5, D-10, D-20, N-0, N-5, N-10, N-20, S-0, S-5, S-10, and S-20. Sandstone sampling information and physical test equipment are shown in Figure 1.

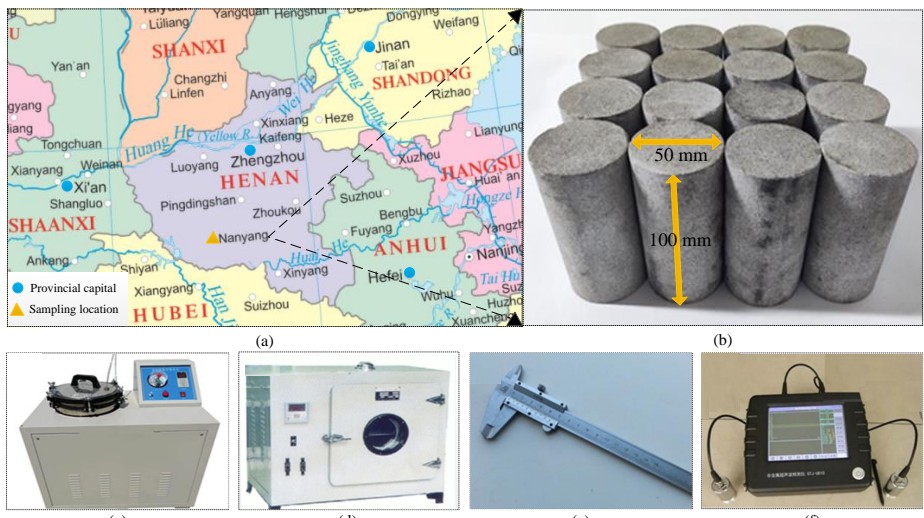

**Figure 1.** Sandstone sampling information and physical test equipment: (**a**) sampling location; (**b**) sandstone samples; (**c**) saturator; (**d**) dry oven; (**e**) saturation instrument; (**f**) acoustic test instrument.

### 2.2. Experimental Equipment and Methods

Dry, natural, and saturated specimens were prepared as follows, dry group: put the test sample into the dry oven for dry, set the dry temperature to 105 °C according to MT 224-1990 "Method for determination of permeability coefficient of coal and rock", take out, weigh, and record it after drying for 48 h, and then put it back into the dry oven. After 24 h, take out and weigh it again, compare the results of the two times [15]. If the quality continues to decline, repeat the above process. If the quality does not change twice, put the rock sample in the dry oven for cooling, and then wax seal it. Natural group: weigh and record after sample preparation, and wax seal until the test is carried out. Saturated group: place the sample in the saturation instrument, fill it with deionized water until the rock sample is completely submerged, seal and cover the vacuum water saturation instrument, pump it to the vacuum state, keep it for 48 h, take it out and weigh it, replace the rock sample after recording, take it out and weigh it again after 24 h, and compare the results of the two times. Repeat the technique outlined above if the quality keeps improving. If the quality does not change, remove any remaining water from the rock sample's surface and wax seal it. Calculate the water content of the sample according to Formula (1)

$$\omega = \frac{m_1}{m_0} - 1 \tag{1}$$

where: $m_1$ is the actual water mass in the rock sample, $m_0$ is the mass of dry rock sample, and the unit is g; $\omega$ is the water content of rock sample, unit is %.

In addition, the wave speed of sandstone samples in dry, natural, and saturated states were measured by acoustic wave tester. The water content, density, and wave speed of sandstone tests are shown in Table 1.

**Table 1.** Water content, density, and wave speed of sandstone.

| Number | Moisture Condition | Water Content (%) | Density (kg/m³) | Wave Speed (km/s) |
|--------|--------------------|-------------------|-----------------|-------------------|
| D | Dry | 0 | 23.553 | 3.378 |
| N | Nature | 1.81 | 23.845 | 3.521 |
| S | Saturated | 3.24 | 24.762 | 3.571 |

Uniaxial and triaxial compression tests were carried out on sandstone samples with different water content using a digitally controlled electro-hydraulic servo testing machine RMT-150B developed by Wuhan Institute of Rock and Soil Mechanics, Chinese Academy of

Sciences. The testing device's maximum axial load is 1000 kN, and its maximum confining pressure is 50 MPa. The displacement-controlled loading method was used during the uniaxial and triaxial compression tests, with the loading rate of 0.002 mm/s, and the rock sample was loaded once until it was destroyed. In the triaxial compression test, three sets of confining pressures—5 MPa, 10 MPa, and 20 MPa—were established. After, it was loaded at 0.5 MPa/s by means of stress loading, loaded to a specific confining pressure and then stabilized. The DS5-8 acoustic emission monitoring device was used to continuously track sandstone's acoustic emission events while it was being loaded. Vaseline was used to secure tight contact between the rock sample and the sensor and prevent signal transmission issues between the acoustic emission probe and the rock sample surface. The acquisition threshold for acoustic emission signals was set at 40 dB, and the sampling frequency was 5 MHz. In Figure 2, the test system is displayed [16,17].

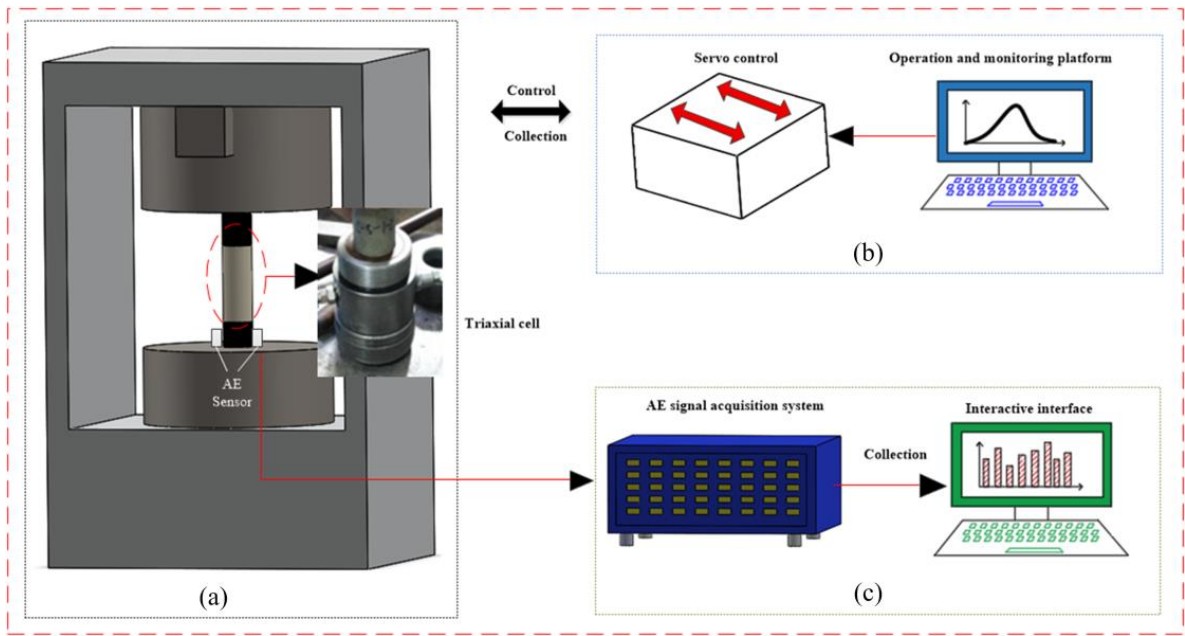

**Figure 2.** Test system: (**a**) RMT-150B rock mechanics test system; (**b**) rock mechanics control system; (**c**) acoustic emission synchronous detection system.

## 2.3. Establishment of Model and Determination of Parameters for Numerical Simulation

EDEM is a high-level discontinuous medium program software based on the discrete element method. It can be used to study the fracture problem of rock-like materials, which are essentially particle aggregates. It can reflect the crack evolution characteristics and failure mechanism of media under stress conditions. One advantage of EDEM is that it can simulate gravity accumulation when particles are accumulated, which makes internal pores generated in the sandstone model, so as to restore sandstone properties to the greatest extent, and it can reflect the compaction stage of sandstone well when simulating uniaxial compression [18,19].

First, a numerical analysis model was generated in the program, which is completely consistent with the size of sandstone samples. The model body was composed of particles, and the edges were constrained by setting "walls". Based on the Hertz Mindlin with Bonding and Hertz Mindlin with JKR particle contact model of EDEM software, the model particles were bonded to generate strength. After completion, the side wall was deleted, and the loading process was simulated by giving the top loading plate motion. The establishment process of the numerical model is shown in Figure 3. The loading speed was the same as the physical test, both of which were 0.001 mm/s.

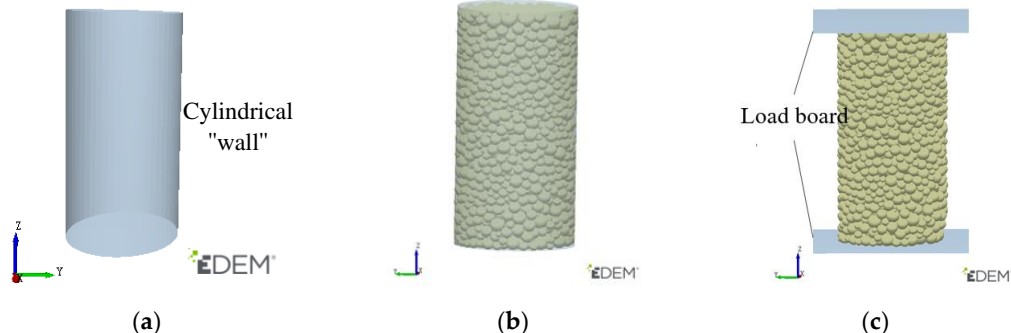

**Figure 3.** Establishment process of numerical model: (**a**) generate surrounding "walls"; (**b**) randomly generate particles; (**c**) calculation model.

Before the loading test, the microscopic parameters of the numerical model were calibrated first. Through the calculation method given by Shen [20], the mechanical parameters such as elastic modulus and Poisson's ratio measured in indoor tests were taken into account, and the mesoscopic parameters of the numerical simulation model were obtained to preliminarily establish the model body. Then, the simulation test was carried out at the same loading rate to obtain the uniaxial compressive strength, elastic modulus, and other results of the numerical model. The "trial and error method" was used to constantly adjust the microscopic parameters of the numerical model until the simulation calculation value was close to the real value obtained from the indoor test. The energy layer was applied to the particle surface to simulate the water content of the rock samples. This paper mainly studies the compressive strength of sandstone and does not discuss the tensile strength of rock. See Table 2 for the calibration results of relevant microscopic parameters.

**Table 2.** Microscopic parameters of sandstone model.

| Number | Normal Contact Stiffness (N/m³) | Tangential Contact Stiffness (N/m³) | Critical Normal Stress (Pa) | Critical Shear Stress (Pa) | Surface Energy (J/m²) |
|---|---|---|---|---|---|
| D | $5.25 \times 10^{12}$ | $5.1 \times 10^{12}$ | $6.8 \times 10^{7}$ | $4.4 \times 10^{7}$ | 0 |
| N | $2 \times 10^{12}$ | $1.85 \times 10^{12}$ | $6.8 \times 10^{7}$ | $4.4 \times 10^{7}$ | 10 |
| S | $1.4 \times 10^{12}$ | $7.2 \times 10^{11}$ | $6.8 \times 10^{7}$ | $4.4 \times 10^{7}$ | 20 |

The numerical calculation method was used to simulate the rock, and the simulation results were mainly judged by the compressive strength and elastic modulus. Taking saturated rock sample as an example, Figure 4c shows the comparison between the stress-strain curves obtained from the test and the EDEM simulation results.

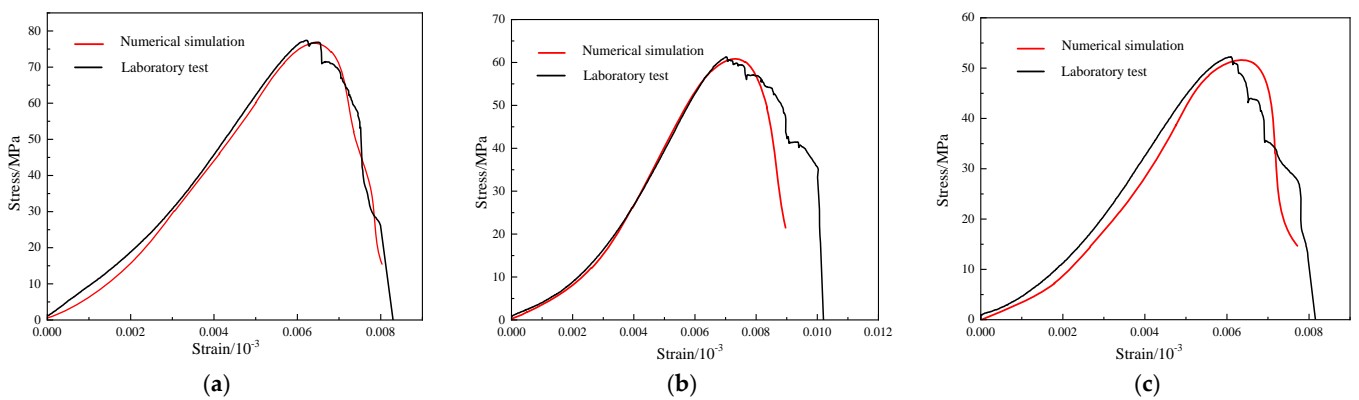

**Figure 4.** Stress-strain curves of sandstone laboratory test and numerical simulation: (**a**) dry sample; (**b**) nature sample; (**c**) saturated sample.

EDEM simulation results show that the peak strength of the material is 52.2 MPa and the elastic modulus is 11.078 GPa. The peak strength of rock material obtained by laboratory test was 51.335 MPa, and the elastic modulus was 10.418 GPa. The results show that there was a deviation of about 1.657% between the laboratory test and numerical simulation, and the difference of elastic modulus was 5.958%, which is within the allowable range.

## 3. Results and Discussion

### 3.1. Effect of Water Content on Stress-Strain Curve of Sandstone under Different Confining Pressures

Compression tests of sandstone under different confining pressure and water content were carried out, and the stress-strain curve is shown in Figure 5. With $\sigma_3 = 0$ MPa as an example, the effect of water content on the sandstone stress-strain curve at various points during loading was analyzed [21,22]. The stress-strain curve of the rock samples was significantly impacted by changes in water content, as illustrated in Figure 5a. The rock samples' peak strength in the dry state was the highest, measuring 77.40 MPa. The compaction stage was shorter, and the elastic stage was longer than that in the natural and saturated states. There was no obvious yielding stage. There was no warning before the failure. The stress after the peak decreased rapidly, and the bearing capacity was lost under a very short strain. The failure had obvious brittleness. The higher the water content, the more obvious the compaction stage of rock sample, the stress-strain curve at the initial stage of loading appeared obviously bending, the elastic stage became shorter, and there was an obvious yield stage. The curve obviously slowed down before reaching the peak, and there were obvious crack signs before failure. Peak rock strength of both natural and saturated rock samples considerably reduced as water content rose. Compared with dry rock sample, the peak strength of natural rock samples was 61.27 MPa, a decrease of 16.13 MPa, a decrease of 26.38%; the peak strength of the saturated rock samples was 52.24 MPa, a decrease of 25.16 MPa and 48.16%. Through comparison, it can be seen that water content significantly reduced the peak strength of sandstone.

The higher water content, the lower triaxial compressive strength of sandstone. When the confining pressure was 5 MPa, the compressive strength of sandstone in dry, natural, and saturated state was 107.07 MPa, 99.18 MPa, and 82.67 MPa, respectively. Compared with dry rock sample, the strength of natural rock samples decreased by nearly 7.96%, and the strength of saturated rock samples decreased by nearly 29.51%; when confining pressure was 10 MPa, the compressive strength of sandstone in dry, natural, and saturated state was 156.94 MPa, 126.62 MPa, and 115.57 MPa, respectively. Compared with dry rock sample, the strength of natural rock samples decreased by nearly 23.95%, and that of saturated rock samples decreased by nearly 35.80%; when confining pressure was 20 MPa, the compressive strength of sandstone in dry, natural, and saturated state was 211.86 MPa, 190.16 MPa, and 185.22 MPa, respectively. Compared with dry rock sample, the strength of natural rock samples decreased by nearly 11.41%, and that of saturated rock samples decreased by nearly 14%. Through comparison, it was found that the increase of water content significantly reduced the compressive strength of sandstone.

Figure 6 shows the fitting relation curve of sandstone peak strength with water content and confining pressure. Figure 6a illustrates that the peak strength of sandstone drops linearly as the water content rises. It can be concluded that confining pressure has little impact on the rate of decline of compressive strength of sandstone based on the influence coefficient of different water contents on the compressive strength of sandstone samples under four distinct types of confining pressure. Figure 6b illustrates that the increase in confining pressure improves the compressive strength of dry, natural, and saturated samples. Excellent linear fitting relationships exist between the compressive strength and confining pressure of samples with various water contents. Under all three circumstances, the fitting relationship coefficient ($R^2$) was higher than 0.9791. Comparing the slopes of the three curves, the slope of the dry state is significantly larger than that of the natural and

saturated state, indicating that the increase of water content will reduce the sensitivity of confining pressure to strength gain.

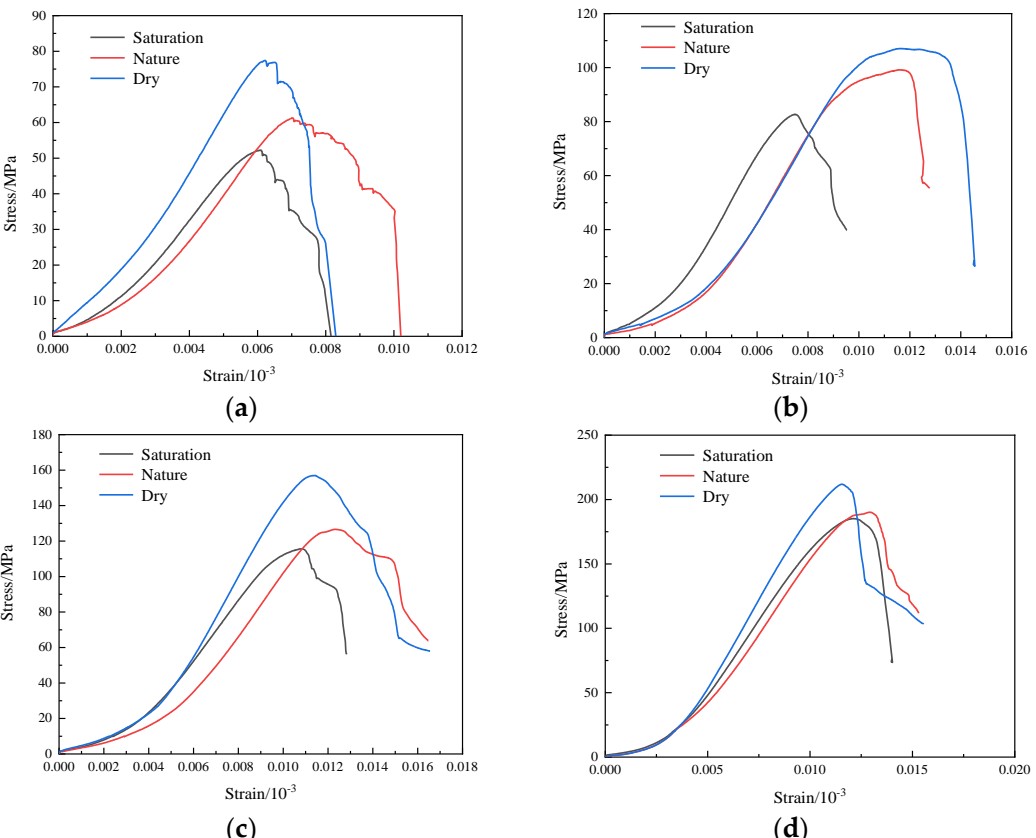

**Figure 5.** Stress-strain curves of sandstone samples with different water content under different confining pressure conditions: (**a**) $\sigma_3 = 0$ MPa (uniaxial compression); (**b**) $\sigma_3 = 5$ MPa; (**c**) $\sigma_3 = 10$ MPa; (**d**) $\sigma_3 = 20$ MPa.

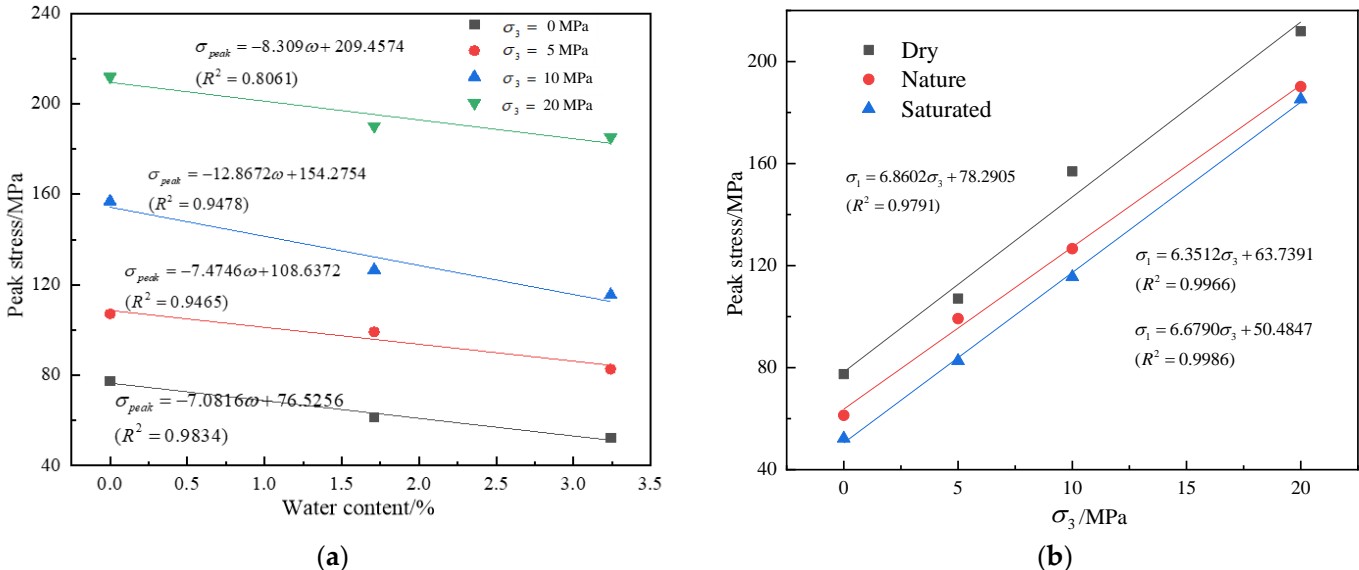

**Figure 6.** Fitting relation curve of sandstone peak strength with water content and confining pressure: (**a**) relationship between peak strength and water content; (**b**) relationship between peak strength and confining pressure.

### 3.2. Analysis of Acoustic Emission Characteristic

The acoustic emission signal of rock is that rock emits sound wave or ultrasonic wave in the process of deformation and fracture [23–25]. Analyzing the relationship between acoustic emission signals and crack propagation characteristics during rock fracture is helpful to study the internal crack evolution of sandstone with different water content under different confining pressures.

### 3.2.1. Ringing Count and Energy Rate Analysis

The stress in rock exceeds the maximum strength of rock, resulting in rock failure. Stress action mode and failure process on fracture surface during rock failure are called failure mechanism. Acoustic emission ringing count is a commonly used acoustic emission characterization parameter, which can reflect the number of energy release events during the formation and expansion of internal cracks in rock samples. The relationship between the stress-strain curve of dry and saturated rock samples and the acoustic emission ringing count accumulative count is shown in Figure 7.

Figure 7 illustrates that the cumulative ringing count curve exhibits an upward concave growth pattern and how the growth rate of the curve gradually rises as the external load continues to rise. At the low stress level, the curve growth rate is low, and at the high stress level, the curve growth rate is high. According to the characteristics of the stress–strain curve, the loading process can be divided into four stages: I compaction stage, II elasticity stage, III yield stage, and IV failure stage. The acoustic emission ring count characteristics of each stage are as follows [26]:

(1) I compaction stage: the primary microfractures inside the rock are closed under external loading, and the elastic strain energy is released due to the occlusion and friction of the particles near the primary and dissolved fractures, which generates a small amount of acoustic emission signal, and with the compaction of the rock samples skeleton in the loading process, the acoustic emission ring count is gradually reduced by the complete closure of the fractures and reaches the lowest at the end of stage I. The number of acoustic emission signals of water-saturated samples is significantly lower than that of dry sample, which is because the fracture is filled with water, the attenuation rate of acoustic emission signal in liquid medium is greater than that in gas medium, and water acts as a lubricant during particle occlusion and friction to reduce acoustic emission signal generation, resulting in the acoustic emission signal intensity in water-saturated condition being lower than that in dry condition. The larger the surrounding pressure, the more significant the effect.

(2) II elastic stage: the rock samples mainly underwent elastic deformation. With the increase of the load, the compacted rock samples gradually entered the linear elastic change stage, and a slight slip appeared between the cracks of the rock samples, but it had not yet reached the crack initiation stress of the rock. The acoustic emission ringing count as a whole remained at a low level, and the slope of the cumulative ringing count curve was basically unchanged; the ringing count of the dry sample was significantly higher than that of the saturated samples, and the effect of compaction pressure was minimal at this stage.

(3) III yield stage: the load stress exceeds the rock's crack initiation stress. At the early stage of this stage, in addition to the expansion of the original crack, a small number of new micro cracks are gradually generated in the samples, resulting in a slight increase in the acoustic emission ringing count. With the loading, new cracks are constantly generated and intersected in the rock samples, resulting in local fracture surfaces. The acoustic emission ringing count starts to increase significantly. With the decrease of water content and the increase of confining pressure, the ringing count value and the slope of the cumulative ringing count curve increase significantly.

(4) IV failure stage: the internal fracture of the rock intensifies, the local fracture on surface starts to penetrate, eventually forming a rupture surface accompanied and by a sharp increase in strong acoustic emission signals, and the peak value of acoustic

emission ringing count appears at this stage. The slope of the cumulative generation curve starts to surge.

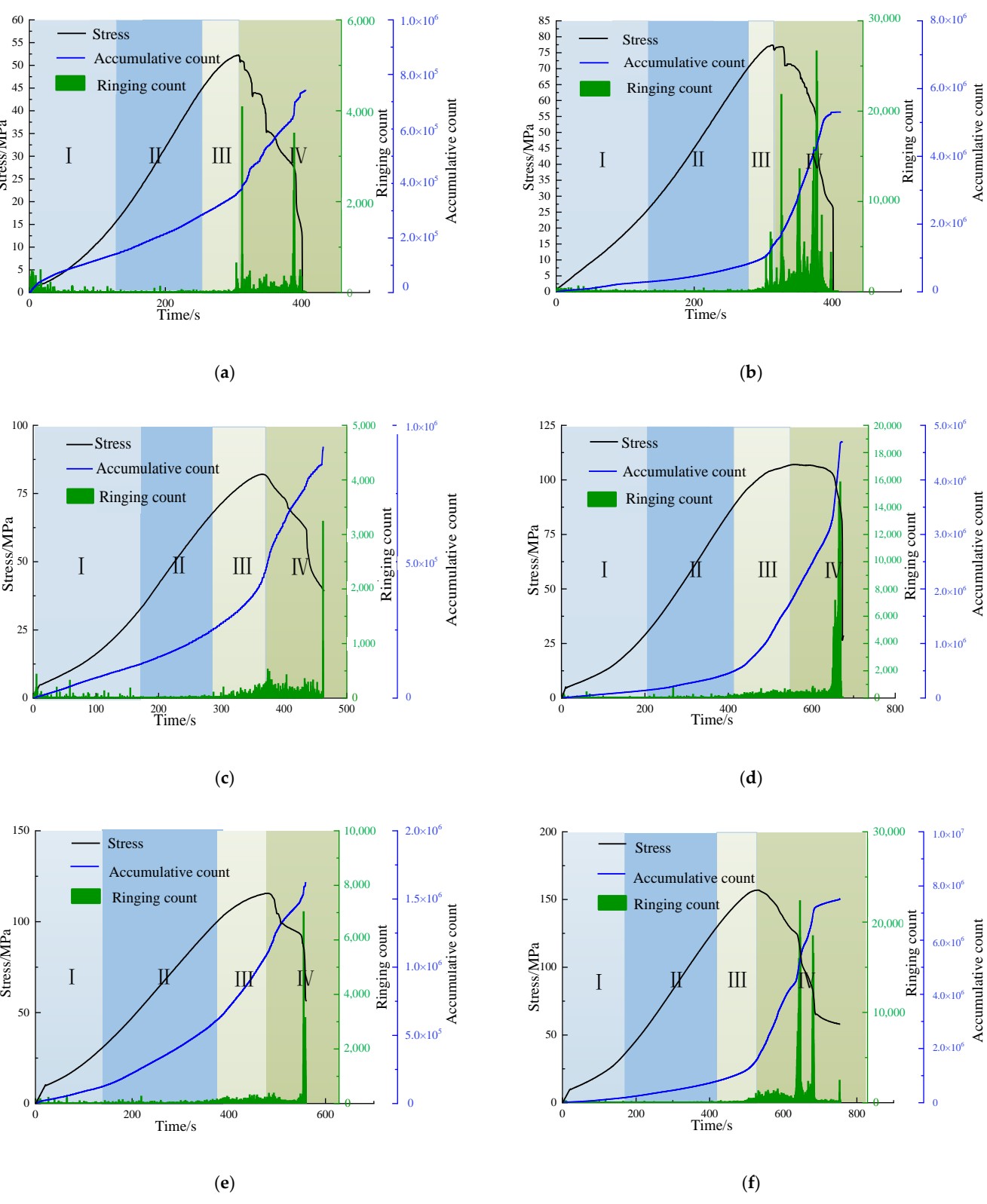

**Figure 7.** *Cont.*

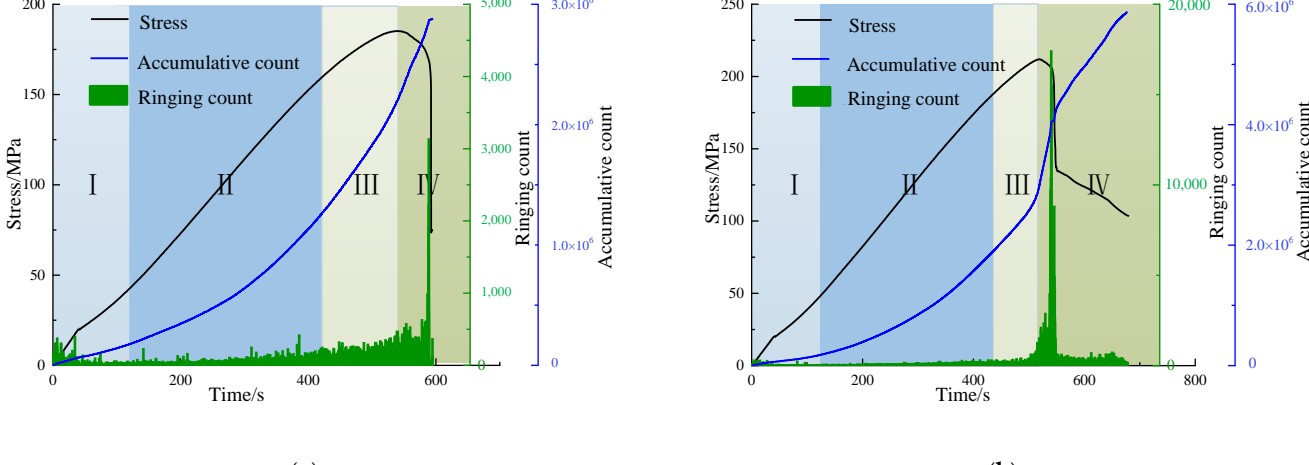

**(g)**   **(h)**

**Figure 7.** Overall stress-strain curves and ringing count of rock samples with different water content: (**a**) S-0; (**b**) D-0; (**c**) S-5; (**d**) D-5; (**e**) S-10; (**f**) D-10; (**g**) S-20; (**h**) D-20.

Comparing the test results of rock samples with different water contents, the following conclusions are obtained:

(1) The acoustic emission ring count of the dry sample is always bigger than that of the saturated sample in each corresponding stage when the confining pressure is the same. The maximum slope of the cumulative AE ringing count curve of the dry sample is always higher than that of the saturated sample, which indicates that the dry sample is more conducive to the generation of acoustic emission signals; another effect of water is reflected in the unstable fluctuation of acoustic emission signal. Under triaxial compression, the acoustic emission signal of water saturated samples fluctuates significantly at the yield stage, while the ringing count of dry sample increases steadily at this stage, which is the same as the research results of Yao [27].

(2) There are obvious differences in the characteristics of acoustic emission ringing count under the two loading modes. The acoustic emission ringing count of the samples under uniaxial compression is mostly concentrated in the failure stage, and less in the pre-peak stage. However, the ringing count of the samples under triaxial compression is also distributed in the yield stage, indicating that the intersection and penetration of cracks and the generation of local fracture surfaces of the samples before reaching the peak strength under uniaxial compression are lower than those under triaxial compression, Under triaxial compression, the energy of rock samples is released violently before the peak strength; at the same time, the slope of the cumulative AE ring count curve under uniaxial compression increased significantly at the peak point, while the point of sharp increase in the slope of the accumulative AE ring count curve under triaxial compression was located in the third stage, and the curve at the peak point was smoother.

AE energy rate is represented by wave carrying energy and duration, which can reflect the degree of fracture in the rock sample. The AE energy rate of the stress curve of sandstone samples with different water content is a comprehensive reflection of the energy and time of the monitored acoustic emission events, which can be used to characterize the degree of micro fracture in sandstone bodies. Overall stress-strain curves and energy rate and accumulative energy rate for rock samples with different water content are shown in Figure 8.

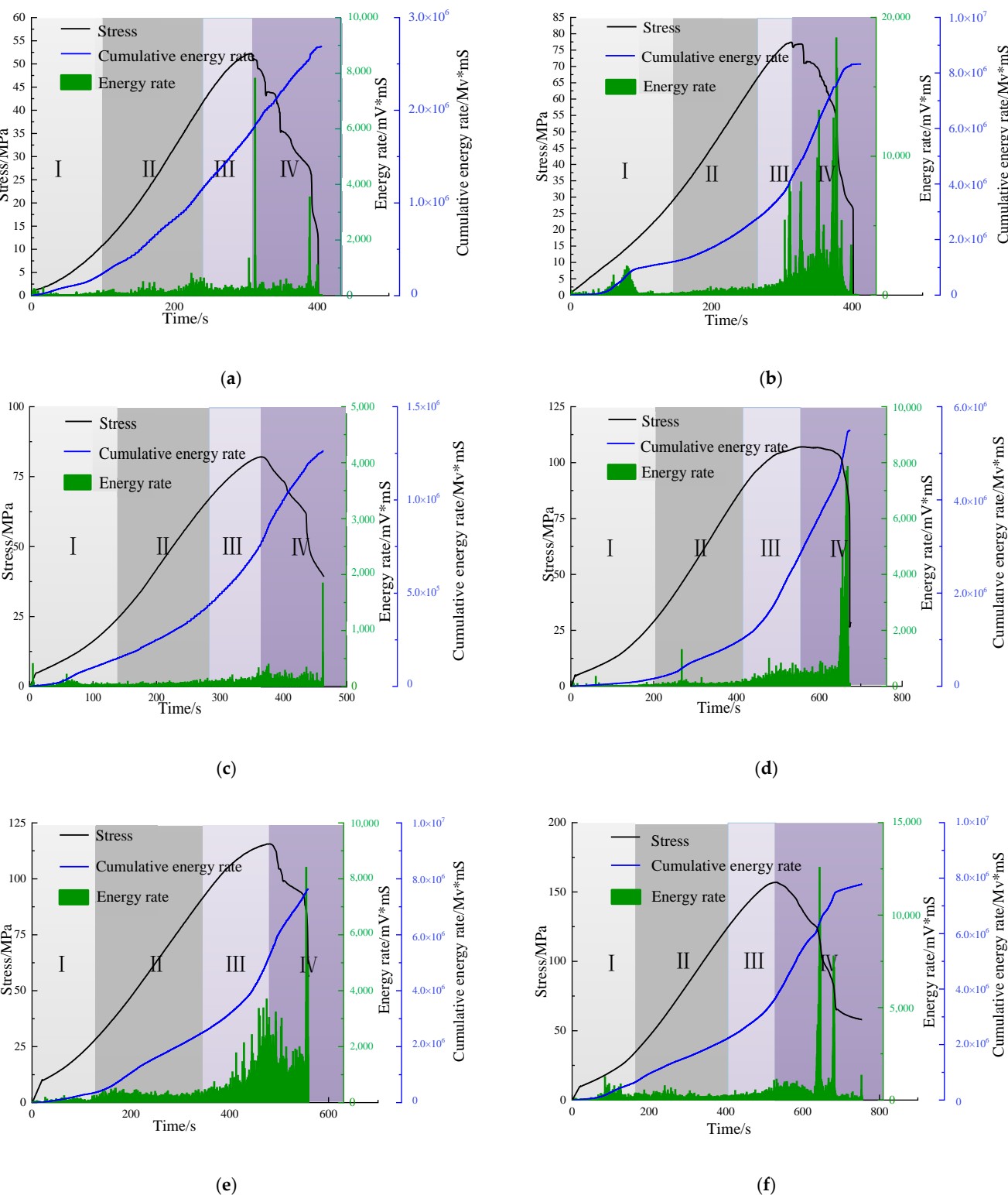

**Figure 8.** *Cont.*

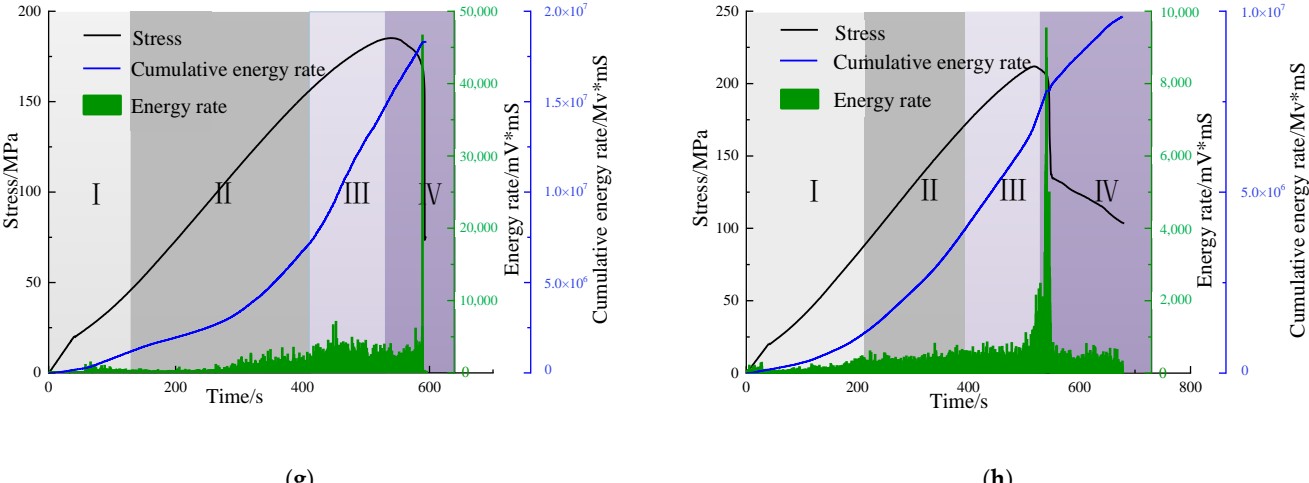

(**g**)                                                                                                    (**h**)

**Figure 8.** Overall stress-strain curves and energy rate, accumulative energy rate for rock samples with different water content: (**a**) S-0; (**b**) D-0; (**c**) S-5; (**d**) D-5; (**e**) S-10; (**f**) D-10; (**g**) S-20; (**h**) D-20.

Comparing Figures 7 and 8 shows that the acoustic emission energy characteristics are similar to the acoustic emission ringing count characteristics, and also have obvious stages. AE energy rate is lower in the compaction stage and elastic stage; at the early stage of the yield stage, the AE energy rate keeps increasing at a low speed in the later yield stage, the AE energy rate increases significantly; in the failure stage, the slope of the cumulative energy rate curve reaches the maximum at this stage. At the same time, there are many AE energy rate events at this stage, and the highest AE energy rate of the samples occurs at this stage.

Different from the distribution characteristics of ringing count, there is no obvious regularity in AE energy rate in the compaction stage, because AE energy rate reflects the energy released by an acoustic emission signal, and ringing count is the number of acoustic emission signals. Therefore, the ringing count in the compaction stage shows a gradual decrease with the loading.

### 3.2.2. Relationship between RA and AF Parameters and Failure Mode

Many researchers grouped the AE data into shear and tensile clusters and used the AE parameters to study the cracking mode of microcracks in experiments. The RA value can be obtained by dividing the rise time by the amplitude, and the unit is MS/v. The AF value is obtained by the ratio of the number of ringing times to the duration, in kHz. In general, when the AF value of acoustic emission signal is low and the RA value is high, shear fractures develop in rock samples; in contrast, tensile cracks are developed in the rock samples, as shown in Figure 9 [28]. According to the crack classification procedure of JCMS-III B5706 specification, it is recommended to use the RA-AF diagram shown in Figure 9 to partition the acoustic emission data to distinguish tensile cracks and shear cracks.

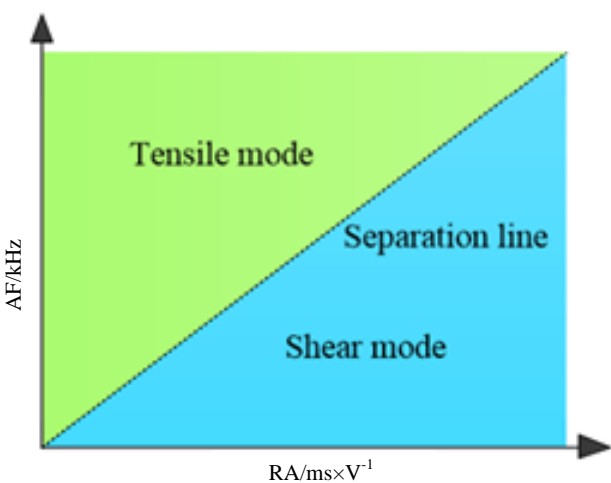

**Figure 9.** Failure mode classification of cracks according to RA-AF.

The Rise time/Amplitude (RA) value and Ring count/Duration (AF) value of sandstone samples is calculated and a scatter plot is drawn, as shown in Figure 10. AF and RA are distributed in [0, 200].

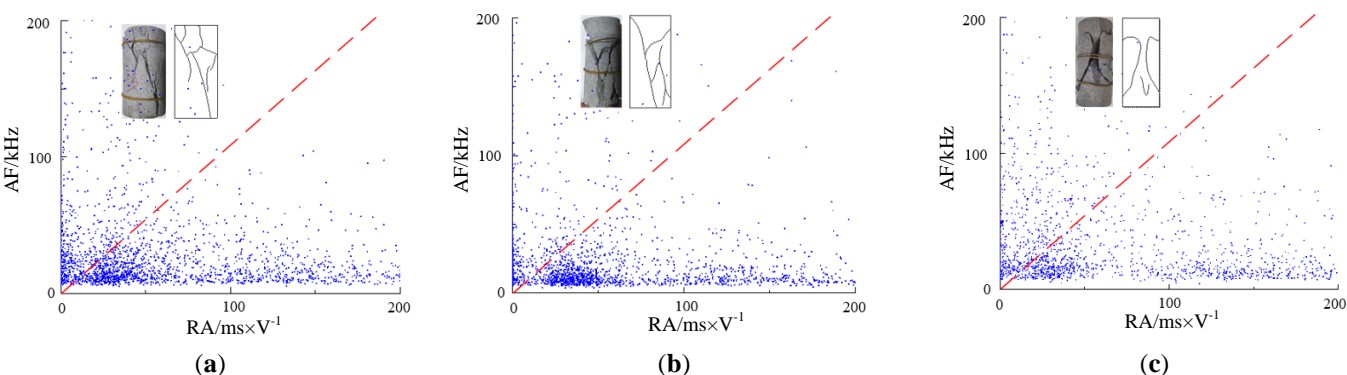

**Figure 10.** RA-AF distribution of different samples under uniaxial compression: (**a**) S-0; (**b**) N-0; (**c**) D-0.

Figure 10 illustrates the development of shear cracks and tensile cracks in the samples in the three states, with shear cracks serving as the predominant type. From the distribution characteristics of scatter points, there is little difference in the distribution range of tensile crack scattered points of samples in the three states, but the high-density distribution range of shear crack signal points of dry sample is larger, and with the decrease of water content, the overall proportion of scattered points below the separation line is gradually increasing. It shows that the lower the water content, the more favorable the formation of shear cracks, the more unfavorable the formation of tensile cracks.

Because there are many overlapping points in the scatter plot, and the skip point method is used in the drawing, in order to verify the above conclusions and further explore the failure mode and crack evolution of rock samples, the complete acoustic emission signals are statistically analyzed, as shown in Figure 11. From the quantity of AE signals, with the decrease of water content, the total number of the two types of cracks decreases, which corresponds to the gradual decline of the final damage and fragmentation of the water saturated rock samples to the dry sample in the test.

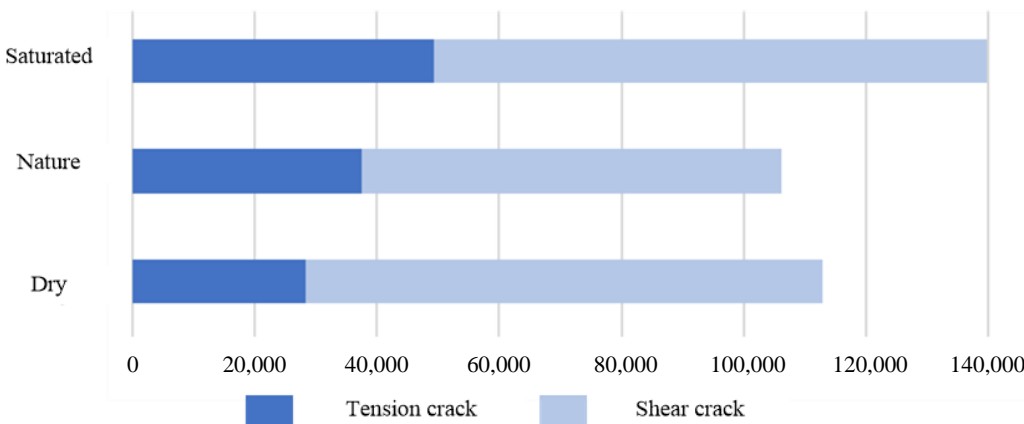

**Figure 11.** Statistics of total signal number of rock sample under different water content.

### 3.3. Influence of Water Content on Sandstone Failure Mode

The failure characteristics of rock samples are the final result of internal microcrack propagation, which contains rich information such as rock sample deformation, force chain evolution path, crack propagation results, etc. Analyzing the failure characteristics of rocks is of great significance for studying the influence of water content on sandstone properties [29].

Figure 12 shows the failure mode of rock samples with different water content. It can be seen from Figure 12 that the final failure mode of the dry sample under uniaxial compression presents a typical "X" conjugate shear failure. The two oblique main fracture surfaces are caused by shear slip and conjugate cross in the middle of the samples. After the failure, the rock samples have good integrity and a small number of fragments, as shown in Figure 12a; with the increase of the water content of the rock samples, the number of cracks increases, and the failure mode changes from the "X" conjugate shear failure in the dry state to the single slope shear failure. In the natural state, the rock samples finally present a "Y" shaped failure mode, and the degree of fragmentation of the rock samples obviously increases, as shown in Figure 12e; as the water content continues to increase, the fractures continue to increase, and the rock samples in the water saturated state finally presents a single slope shear failure mode, with the highest degree of fragmentation, as shown in Figure 12i. With the increase of water content, the ultimate fracture degree of sandstone increases, which is due to the role of water in weakening the cementation ability of rock particles. The uniaxial compression failure mode of rock samples in the three states is shear failure. The influence of water content on the failure mode of rock samples deduced from the final failure mode of rock samples is consistent with the change rule of water content on the failure mode of rock samples judged by RA-AF value in Section 3.2.2. Under the same water content, with the increase of confining pressure, the integrity of the rock samples is higher when it is destroyed, and the final failure form is shear failure, which indicates that confining pressure has an inhibitory effect on the failure of the rock samples.

### 3.4. Analysis of Numerical Simulation Results

Figure 13 is a comparison of the final failure characteristics of sandstone with different water content under EDEM simulation uniaxial compression and the final failure morphology of laboratory tests. When the dry sample is damaged (Figure 13a), the crack is a penetrating shear crack, and the failure form is shear failure; when the rock samples is damaged in the natural sample (Figure 13b), there are two vertical tensile cracks in the rock samples. At this time, the failure mode of the rock mass changes from shear failure to tensile failure; the saturated rock sample is damaged (Figure 13c). At this time, the tensile cracks of the samples are fully developed. The final failure form is typical tensile failure, and the degree of rock fragmentation is even greater than the former. The failure modes of

the above rock samples under different water bearing conditions are consistent with the indoor test results.

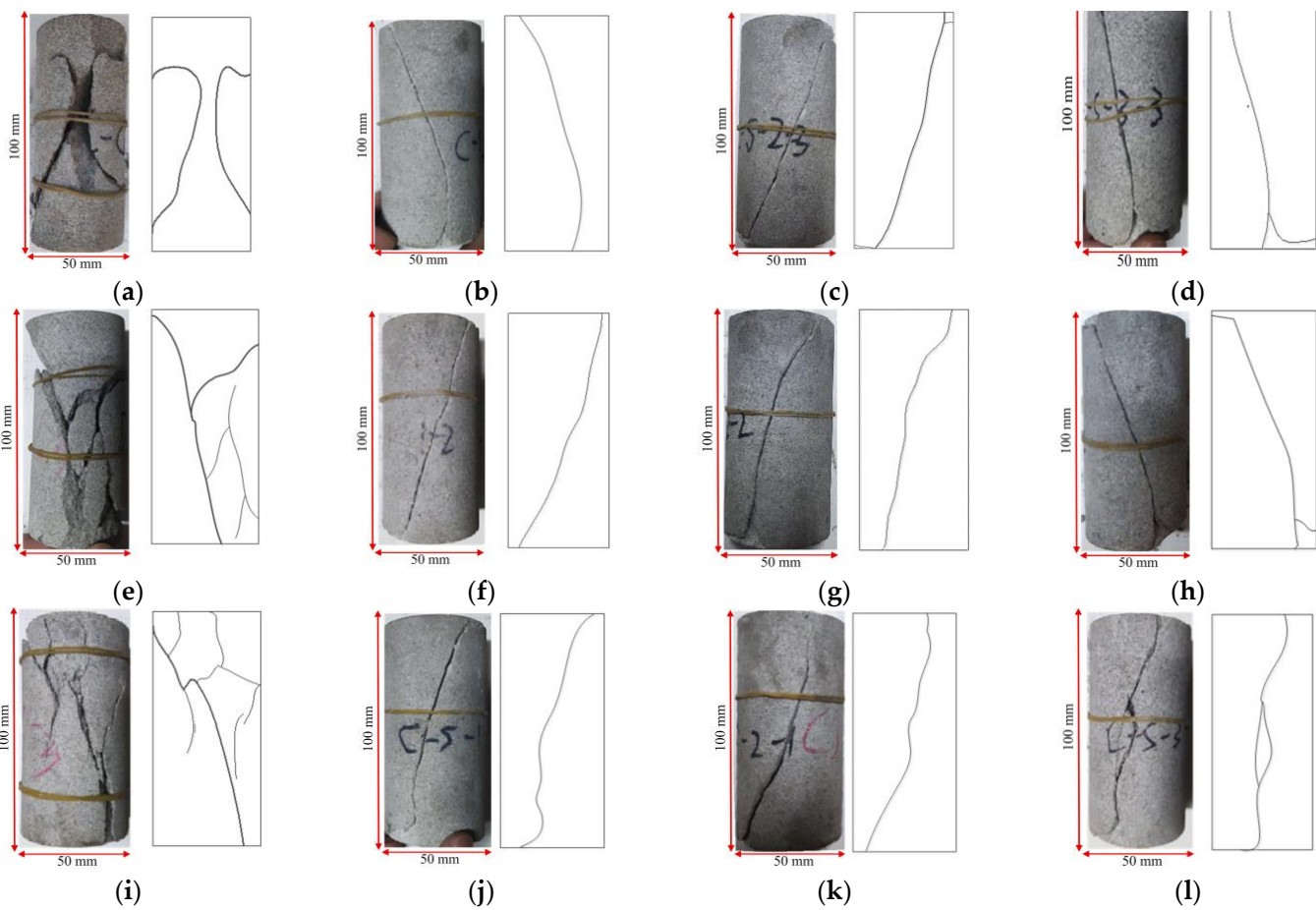

**Figure 12.** Failure modes of sandstone with different water content under different confining pressure: (**a**) D-0; (**b**) D-5; (**c**) D-10; (**d**) D-20; (**e**) N-0; (**f**) N-5; (**g**) N-10; (**h**) N-20; (**i**) S-0; (**j**) S-5; (**k**) S-10; (**l**) S-20.

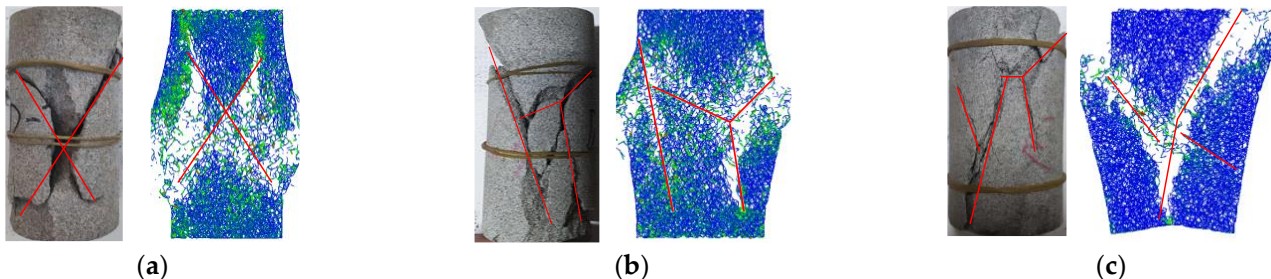

**Figure 13.** Uniaxial compression test results and simulation comparison of sandstone with different water content: (**a**) D-0; (**b**) N-0; (**c**) S-0.

Table 3 shows the crack propagation process of rock samples in different loading states. The stress distribution nephogram of bond at the boundary point from compaction stage to elastic stage, yield point, and peak point is shown in the table. By comparison, the bond stress of the three states rock samples at each stage point increases with the decrease of water content. Compared with other rock samples, the stress of the bond in the saturated rock sample is at a low level during the whole loading process. The cloud map is evenly blue, and a small part of the crack propagation zone has uneven stress. The cloud picture of the dry sample is uniform green during loading, and the bonding stress is always at a high level. The non-destructive areas at both ends of the rock samples are blue, and the stress is

slightly lower. Table 3a,d,g rock samples are at boundary point from compaction stage to elastic stage, compared with the initial stress state, the bond of the rock samples is overall tighter without fracture; Table 3b,e,h is the yield point of the rock samples. The stress concentration occurs inside the rock samples, and the bond has been obviously connected, which corresponds to the internal cracks in the laboratory test. Table 3c,f,i is at the peak point of the rock samples. At this time, the broken bond has been connected, many cracks have appeared in the cloud map, and the edge has also fallen off, and the bearing capacity of the rock samples began to decline. Comparison Table 3a,d,g, compaction stage and elastic stage transition point, with the decrease of water content, the stress cloud of rock samples is more uniform, but the stress level increased significantly. At the peak point, with the increase of water content, the internal crack path of rock samples increases obviously.

**Table 3.** Bond stress nephogram of rock samples under different state.

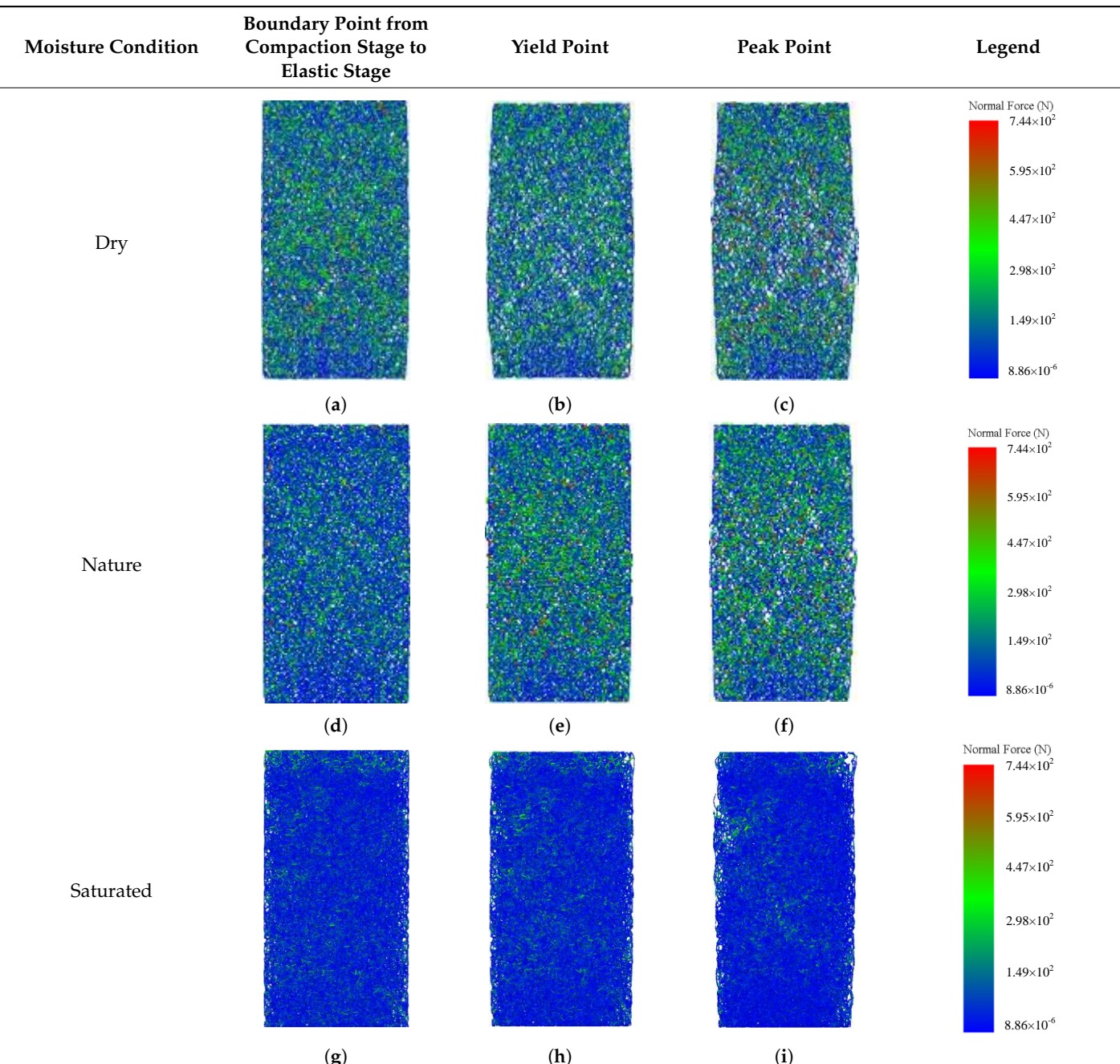

| Moisture Condition | Boundary Point from Compaction Stage to Elastic Stage | Yield Point | Peak Point | Legend |
|---|---|---|---|---|
| Dry | (a) | (b) | (c) | |
| Nature | (d) | (e) | (f) | |
| Saturated | (g) | (h) | (i) | |

Figure 14 shows the relationship between the amount of damage and the water content state during bond formation. It can be seen from the figure that with the increase of water content, the number of damaged bonds continues to rise. With the total number of particles unchanged, the total number of bonds originally generated was 97,572, and the number of bonds destroyed in the dry state was 34,076; the number of bonds broken in natural state is 38,396; the number of broken bonds in the saturated state is 44,636. With the increase of the water content, the number of broken bonds increases synchronously, which is the reason for the strength decline of the rock samples in the later loading process. This also corresponds well to the water eroding the rock mass inside the rock samples and reducing the adhesion between particles.

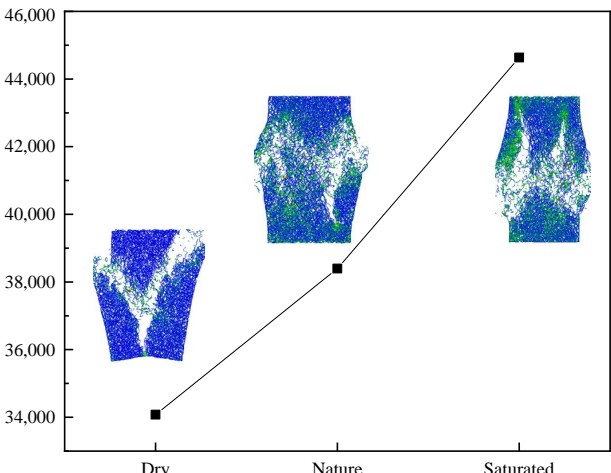

**Figure 14.** Relationship between the amount of damage bond and water content.

Figure 15 shows the relationship between the number of broken bonds and the stress-strain curve of sandstone under different water content. It can be seen from the diagram that the bonding of the three states did not break at the initial stage of loading (stage I); in the elastic stage (stage II), the fracture began to occur, and the number growth was stable and slow. With the continuous compression, the rock samples enter the pre-peak stage (stage III), the number of bond failure in the three states increases significantly, and the slope of the curve increases significantly. Eventually, the rock samples enters the failure stage (stage IV) and the number of broken bonds begins to increase dramatically, peaking at complete failure The time when the bond fracture first appears in the three different states is gradually advanced with the increase of water content, and the rising slope of the corresponding curves of the three states also decreases with the increase of water content after reaching the peak strength, which proves that water content is closely related to the weakening of sandstone strength and internal bond.

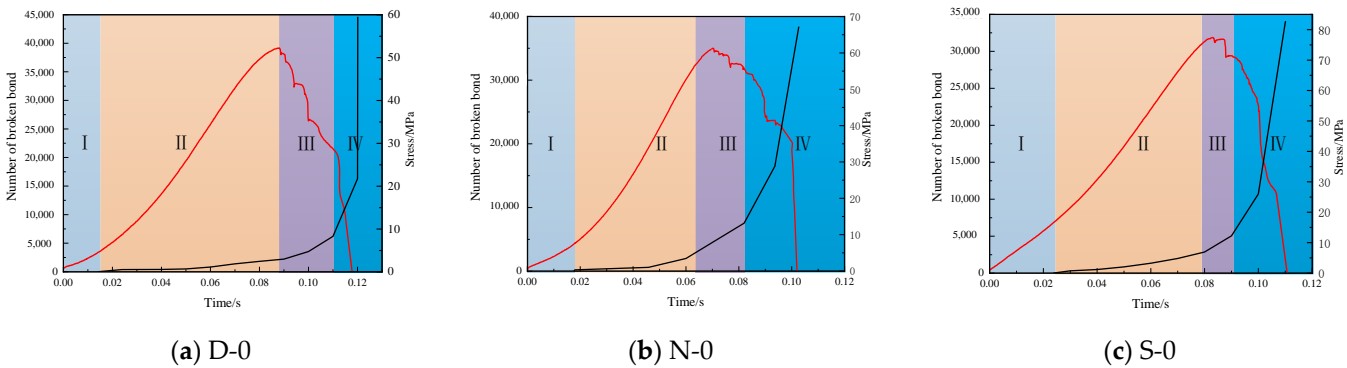

**Figure 15.** Relationship between bond quantity and time of sandstone failure in different states: (**a**) D-0; (**b**) N-0; (**c**) S-0.

## 4. Conclusions

In this paper, the conventional uniaxial and triaxial tests of sandstone with different water content and confining pressures were carried out, and the laboratory tests are simulated by discrete element software to further study internal fracture evolution process of sandstone. The following conclusions are obtained:

(1) Compared with uniaxial compression results, the peak strength of sandstone decreases with the increase of water content. Water action will cause certain damage to the interior of sandstone samples, weaken the cementation between rock particles, and reduce the compressive strength and elastic modulus of sandstone samples. The higher the water content, the more complex the fracture development of sandstone samples, and the more broken the rock samples is. Therefore, water–rock interaction has a great influence on rock strength.

(2) The AE energy rate and ringing count of the rock samples during loading process show obvious "quiet period" to "frequent period" phased changes, and the stage boundary corresponds to the peak strength. With the decrease of the water content of sandstone samples, the proportion of frequent period decreases, and the maximum energy rate increases significantly. Acoustic emission is active in yield stage and failure stage, and a large amount of strain energy and acoustic emission signals are released from rock samples; with the increase of water content, the frequency of rock samples is obviously reduced, and the strain energy released by rock samples under load is reduced. These have reference significance for practical engineering risk prediction.

(3) Using discrete element method to analyze the destruction process of rock samples in detail, the increase of water content inside the rock mass is not conducive to the generation and maintenance of the number of connection bonds in the model body, and the number of destroyed bonds in dry sample at the same loading time is significantly less than that in water-saturated rock samples. It can be seen that the strength of sandstone increases with the decrease of water content. From the micro level analysis, water reduces the connection effect inside the rock mass, and this effect increases with the increase of water content. This result provides a new idea for the study of water–rock interaction in underground engineering by the discrete element method.

**Author Contributions:** Conceptualization, X.H. and Y.L.; methodology, T.W. and J.G; software, Y.L.; validation, T.W. and Y.L.; formal analysis, T.W. and J.G.; investigation, J.G.; resources, Y.L.; data curation, X.H. and T.W.; writing—original draft preparation, Y.L. and J.G.; writing—review and editing, T.W. and X.H.; supervision, X.H.; project administration, J.G.; funding acquisition, J.G. and X.H. All authors have read and agreed to the published version of the manuscript.

**Funding:** "This research was funded by the National Natural Science Foundation of China, grant number 52178388" and "The Opening Project of Key Laboratory of Highway Bridge and Tunnel of Shaanxi Province (Chang'an University), grant number 300102211517" and "The Key Scientific and Technological Project of Henan Province, grant number 212102310292" and "The Fundamental Research Funds for the Universities of Henan Province, grant number NSFRF210337".

**Institutional Review Board Statement:** Not applicable.

**Informed Consent Statement:** Not applicable.

**Data Availability Statement:** Not applicable.

**Conflicts of Interest:** The authors declare no conflict of interest.

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
