# Peer review of "Study on the Influence of Water Content on Mechanical Properties and Acoustic Emission Characteristics of Sandstone: Case Study from China Based on a Sandstone from the Nanyang Area"

_sustainability, doi:10.3390/su15010552_

Round 1

Reviewer 1 Report

The ms is generally well organized with good data to be published. The following are concern to improve the quality. 

1. Abstract . Introduce where are the sample from and their basic rock compositions. And also how many samples you tested, and the moisture content or ranges used, as well as the measurement condition, like confining pressure and mechanical route.

Implicate the meaning of this study at the end of the abstract. 

2. In the second paragraph of the Introduction, you listed too many names, while the authors still cannot know the key advances. Please list the scientific issues, then following by key authors work. 

3. The figure 1-a is unclear and also seem unnecessary, you can use a simple  north China figure. 

Scale bar is necessary for the subfigure f. 

4. Section 5.1should be moved to the methods section. 

5. Revise the section titles as indicated in the annotated pdf.

6. Concise the conclusions and implicate the signifcances. 

7. Add scale bar for the Fig. 10 .

Author Response

Dear reviewer

Thank you for your suggestions on the manuscript. We have revised it according to your suggestions. See the attachment for details.

Reviewer 2 Report

The reviewer thanks the authors and editors for the opportunity to review the manuscript.

This article discusses the issues of the influence of water content on the mechanical properties of sandstone from the Nanyang region (China). Based on the study, it is shown that the water content in the stone structure significantly weakens the stone.

Specific comments:

Title: The article refers to only one sandstone from China. The title should be clarified by adding "case study from China based on a sandstone from the Nanyang area".

Abstract: The authors do not state the purpose and rationale of the research conducted.

Introduction:

1) There is a lack of information in the introduction what is the purpose of the article? What are the practical and theoretical aspects of this article? Why did the authors undertake this topic? Why do the authors only study sandstones? What new things do the authors propose from the scientific work already described?

2) The authors refer mainly to papers by Chinese authors. The literature review should be supplemented with European or American publications.

Section 2: Subsection numbers should be corrected.

Section 2, line 90: it should be specified that the samples are from China.

Section 2 (for example, line 90, 110): The authors use "standards", "specifications". Which documents exactly are they referring to? The authors should cite the number of the standard.

Section 2.1: Were the sandstone samples for testing only from one site? It seems that samples from a single site do not provide adequate knowledge in the scope that the authors have undertaken. If the authors are conducting research on one material then this should be included in the title as a case study.

Author Response

(The authors gave the same response as above.)

Round 2

Reviewer 2 Report

All my comments were considered and corrections were done in the manuscript. 

I recommend the manuscript for publishing.